# Neurobiological Underpinnings of Hyperarousal in Depression: A Comprehensive Review

**DOI:** 10.3390/brainsci14010050

**Published:** 2024-01-04

**Authors:** Musi Xie, Ying Huang, Wendan Cai, Bingqi Zhang, Haonan Huang, Qingwei Li, Pengmin Qin, Junrong Han

**Affiliations:** 1Key Laboratory of Brain, Cognition and Education Sciences, Ministry of Education, School of Psychology, Center for Studies of Psychological Application, Guangdong Key Laboratory of Mental Health and Cognitive Science, South China Normal University, Guangzhou 510631, China; musi_xie@m.scnu.edu.cn (M.X.); ying_huang@m.scnu.edu.cn (Y.H.); 2Key Laboratory of Brain, Cognition and Education Sciences, Ministry of Education, Institute for Brain Research and Rehabilitation, Guangdong Key Laboratory of Mental Health and Cognitive Science, South China Normal University, Guangzhou 510631, China; caiwendan@m.scnu.edu.cn (W.C.); zhangbingqi@m.scnu.edu.cn (B.Z.); haonan_huang@m.scnu.edu.cn (H.H.); 3Department of Psychiatry, Tongji Hospital, Tongji University School of Medicine, Shanghai 200065, China; lianocd@tongji.edu.cn; 4Pazhou Laboratory, Guangzhou 510330, China

**Keywords:** depression, arousal, thalamocortical connectivity, thalamus, EEG vigilance

## Abstract

Patients with major depressive disorder (MDD) exhibit an abnormal physiological arousal pattern known as hyperarousal, which may contribute to their depressive symptoms. However, the neurobiological mechanisms linking this abnormal arousal to depressive symptoms are not yet fully understood. In this review, we summarize the physiological and neural features of arousal, and review the literature indicating abnormal arousal in depressed patients. Evidence suggests that a hyperarousal state in depression is characterized by abnormalities in sleep behavior, physiological (e.g., heart rate, skin conductance, pupil diameter) and electroencephalography (EEG) features, and altered activity in subcortical (e.g., hypothalamus and locus coeruleus) and cortical regions. While recent studies highlight the importance of subcortical–cortical interactions in arousal, few have explored the relationship between subcortical–cortical interactions and hyperarousal in depressed patients. This gap limits our understanding of the neural mechanism through which hyperarousal affects depressive symptoms, which involves various cognitive processes and the cerebral cortex. Based on the current literature, we propose that the hyperconnectivity in the thalamocortical circuit may contribute to both the hyperarousal pattern and depressive symptoms. Future research should investigate the relationship between thalamocortical connections and abnormal arousal in depression, and explore its implications for non-invasive treatments for depression.

## 1. Introduction

Major depressive disorder (MDD), a prevalent neuropsychiatric disorder, is emerging as a serious public health concern. Despite decades of research, the neurobiological underpinnings of depression remain elusive. This may partly be due to incomprehensive investigations regarding the neural basis of arousal—key features of the brain states—in depression [1]. A theory in affective disorders suggests that hyperstable arousal regulation may lead to depressive symptoms such as social withdrawal and sensation avoidance [2,3,4]. In fact, recent evidence supports this abnormal arousal pattern in depression. Specifically, a higher and hyperstable arousal state has been observed in depressed patients compared with healthy, non-depressed individuals [4,5,6]. Moreover, this abnormal arousal pattern is associated with the severity of depressive symptoms and prolonged sleep onset latency in depression [3,7,8]. Therefore, neural mechanisms of physiological arousal appear to play a critical role in depressed patients, affecting their depressive symptoms.

Physiological arousal is considered as a key component of consciousness [9,10], and is supported by complex cooperation between the subcortex and cortex, particularly thalamocortical circuitry [11,12]. Previous studies have demonstrated that arousal affects perception [13], decision making [14], spatial memory [15], and attention [16], and is also a critical characteristic in mental disorders [3,17,18,19,20,21], including depression [3]. For now, altered activities of several arousal neural correlates have been identified in depression, such as the hypothalamic–pituitary–adrenal (HPA) axis [22,23] and locus coeruleus (LC) [3,24]. However, these subcortical findings did not reveal the subcortical–cortical interactions, and could not explicitly explain dysfunctions in high-order cognition or symptoms in depression. Currently, alterations in thalamocortical interactions that underlie the aberrations in physiological arousal in depression remain largely unclear.

In this review, we first present an overview of the behavioral and physiological characteristics of arousal. Next, we provide a comprehensive review of abnormal physiological arousal in depression, including both behavior and psychophysiological evidence. Subsequently, upon overviewing the role of thalamocortical circuits in arousal and MDD, we propose that abnormalities of these circuits could be a key neural mechanism underlying both hyperarousal and depressive symptoms. Finally, we discuss the future outlook for investigating the abnormal arousal in depression, fostering future research on the theoretical understanding of the pathology of MDD and its treatment approaches, including repetitive transcranial magnetic stimulation (rTMS), music interventions, and pharmacotherapy.

## 2. Overview of Physiological Arousal

Arousal is closely linked to consciousness, cognition, and mental disorders [9,25]. The level of arousal influences the performances of various cognitive tasks. For instance, a brief increase in arousal can shorten reaction time in decision making [14], while the restoration of arousal after light sleep improves detection in visuomotor tasks [26]. Additionally, the level of arousal before stimulus can predict perceptual task performance [27,28]. In the context of mental disorders, abnormal arousal is commonly found. Depression, for example, often co-occurs with hyperarousal and sustained tension [3,4,5,29]. Individuals with autism spectrum disorder (ASD) demonstrate abnormal arousal, but arguments exist regarding whether hypoarousal or hyperarousal accounts for the attentional and social skills in autism [21,30]. Therefore, understanding the neural basis of arousal is essential for disclosing the neural mechanisms of consciousness, cognition, and mental disorders.

It should be noticed that, although physiological arousal and emotional arousal have overlapping neural underpinnings, they are distinct concepts [31]. Emotional arousal is related to an individual’s brain and bodily responses to arousing stimuli, focusing on emotional reactivity [32]. On the other hand, physiological arousal measures the degree of wakefulness of the individuals. In this review, we focus on physiological arousal, hereafter referred to simply as arousal.

### 2.1. Physiological Correlates of Arousal

Physiological indices provide a comprehensive evaluation of arousal, providing objective and quantifiable measurements of arousal levels. These indices include behavior, electrodermal activity (EDA), heart rate variability (HRV), and pupil diameter, each of which provides unique insights into arousal states [11,33,34,35,36]. Behavior, as the most intuitive among these indices, visibly changes with shifts in arousal levels, such as when an individual awakens from sleep, performs limb movements, or speaks [11,33]; EDA reflects autonomic nervous system activity, which typically increases during states of high arousal [36]; HRV measures the natural variability in time intervals between heartbeats, and shows an inverse relationship with arousal levels [34]. Additionally, pupil diameter, influenced by the sympatho-vagal balance within the autonomic nervous system, expands with increased physiological arousal [35,37]. Together, these indices provide a solid foundation for assessing physiological arousal levels as well as exploring their neurophysiological underpinnings.

### 2.2. Neural Correlates of Physiological Arousal

The complex interplay of subcortical brain regions is essential for the neural mechanisms of arousal. These regions, primarily comprising the brainstem, hypothalamus, basal forebrain, and thalamus, work in concert to regulate various aspects of arousal, ranging from hormonal responses to neuronal activity [11,12]. This intricate system coordinates the physiological processes underlying our ability to wake and remain alert, influencing cognitive functions and mood states. The brainstem, particularly critical for physiological arousal regulation, contains structures like the reticular formation and various nuclei that are integral to the sleep–wake cycle [38,39,40]. The reticular formation enhances arousal by releasing acetylcholine, while brainstem nuclei, including the locus coeruleus (LC) with its norepinephrine-producing neurons, regulate cortical activities affecting mood and cognitive functions [41]. The ascending reticular activating system (ARAS) from the brainstem, projecting through the thalamus to the cerebral cortex, plays a crucial role in cortical activation and maintaining alertness [42,43,44]. The hypothalamus, which governs the hypothalamic–pituitary–adrenal (HPA) axis, significantly influences arousal [45,46]. Hyperactivity in the HPA axis can lead to increased arousal and stress responses, potentially resulting in anxiety and depression [47]. Within the hypothalamus, the lateral hypothalamus (LH) is critical for wakefulness, with neurons like hypocretin/orexin (Hcrt), glutamatergic, and GABAergic types modulating sleep stages [48,49,50,51]. The basal forebrain (BF) also contributes to arousal, with its cholinergic neurons enhancing cortical activity and wakefulness, and GABAergic neurons promoting arousal by modulating cortical inhibitory interneurons [52,53]. The intricate neuronal dynamics within the BF have significant impacts on cognitive functions and the sleep–wake cycle [52,54,55,56]. Additionally, the thalamus has long been involved in the regulation of sleep–wake cycle and arousal, serving a critical role in the transmission and integration of information [57,58,59]. It primarily relays sensory inputs to the cortex, which is a process essential for perception, emotion, and consciousness [57,60,61], and influences brain activation and consciousness states, primarily through neurotransmitters such as glutamate [61].

Numerous studies have focused on how subcortical brain regions regulate arousal. However, recent research has also revealed that the cortex is also involved in arousal alteration. For instance, the amplitude of global signals (GS), predominantly constituted by activities in primary sensory areas such as the sensorimotor cortex [62,63], negatively correlates with arousal levels [64,65], and increases during light sleep and mild anesthesia [66,67]. This indicates a broader engagement of cortical areas in arousal regulation. Notably, decreased physiological arousal is associated with increased thalamic activity and reduced activity in cortical areas, especially the default mode network (DMN) [68]. During anesthesia-induced unconsciousness, while cortico-cortical functional connectivity is preserved, thalamocortical connectivity is disrupted, with consciousness recovery linked to its restoration [69]. Additionally, the sensorimotor cortex drives dynamic functional connectivity of spontaneous signals across the cortex, and is closely related to arousal regulation (Figure 1A) [70,71,72]. Transient increases in GS, co-occurring with decreased activity in the dorsal midline thalamus, nucleus basalis and midbrain, suggest brief decreases in arousal [63]. Moreover, electrical stimulation of central thalamic regions, such as the central lateral nucleus of the thalamus, can induce widespread cortical activity and elevate arousal levels in animals under anesthesia and sleep states [59,73]. Our previous studies exploring altered arousal from eyes-open to eyes-closed states revealed that interactions between cortical networks are closely linked to arousal, underscoring the cortical contribution to arousal regulation (Figure 1B) [74,75]. These studies demonstrate that arousal regulation is governed by not only subcortical mechanisms, but also involves significant cortical contributions, particularly from areas like the DMN and sensorimotor cortex. 

Taken together, physiological arousal is a complex and dynamic neurological process that engages multiple brain functional regions, including the brainstem, hypothalamus, basal forebrain, thalamus, and cerebral cortex. While initial studies of arousal focused extensively on the role of subcortical regions and hormonal interactions in arousal regulation, recent findings have shed light on the impact of cortical activities and their interaction with the subcortex. These findings indicate that arousal is not solely governed by subcortical mechanisms, but also involves critical contributions from cortical areas, especially through the integration and processing of information relayed by the thalamus. Therefore, the connectivity between cortical and subcortical regions may be a critical mechanism for the abnormal arousal in mental disorders, such as depression.

## 3. Abnormal Arousal in Depression

In this study, we mainly focus on patients (adults) diagnosed with MDD to illustrate the abnormal arousal in depression. We conducted a comprehensive search in the Web of Science for articles published up to November 2023 with the following terms: “MDD AND (“heart rate” OR “heart rate variability” OR “pupil” OR “skin conductance” OR “electrodermal activity”); “MDD AND HPA”; “MDD AND locus coeruleus AND norepinephrine”; and “MDD AND EEG vigilance”. Articles resulting from these searches and relevant references cited in those articles were thoroughly reviewed for this research, and this selection was also completed by searches in the authors’ personal files, where articles published in English were included. 

### 3.1. Behavior Characteristics

Abnormal arousal in depression is manifested in sleep behavior. The DSM-5 has indicated insomnia as a primary symptoms of MDD [7,76]. Numerous studies have revealed that MDD patients suffer from various sleep disturbances. These disturbances include permanently increased inner tension [77], difficulties to relax or to initiate sleep, prolonged sleep onset latency, early morning awakenings with an inability to return to sleep, decreased sleep efficiency, and overall reduced sleep duration [78]. Additionally, electroencephalography (EEG) recordings also revealed irregular sleep architectures in depression that are characterized by a decrease in slow wave sleep and alterations in rapid eye movement (REM) sleep patterns. Specifically, the first REM stage occurs earlier and lasts 3–4 times longer in MDD patients compared to healthy individuals. There is also an increased proportion of REM sleep in the early part of sleep and higher REM density (i.e., more ocular movements during REM sleep) [8].

### 3.2. Physiological Evidence

Physiological evidence further confirms a hyperarousal state in depression that is characterized by several distinct markers. These include elevated heart rate (HR), decreased heart rate variability (HRV), increased skin conductance, larger pupil diameters, hyperactivity of the HPA axis and locus coeruleus–norepinephrine (LC–NE) system, and hyperstable arousal regulation as indexed by EEG vigilance.

#### 3.2.1. Autonomic Function Indices

Abnormal arousal in depression is reflected by autonomic function markers such as HR, HRV, electrodermal activity (EDA), and pupil diameters. Studies have demonstrated elevated HR and decreased HRV in MDD patients at rest [79,80,81,82,83,84,85,86,87]. Specifically, studies controlling for factors such as age, gender, smoking habits, and education levels found that unmedicated MDD patients displayed a higher average HR than healthy individuals [84,86]. Additionally, reviews have consistently shown lower HRV in MDD than healthy individuals [88,89], despite some inconsistent results [87]. A comprehensive study using several autonomic function indices revealed that unmedicated MDD patients in a resting state (20 min in a supine position) exhibited higher HR, increased skin conductance levels and fluctuations, and larger pupil diameters than healthy individuals [87]. These findings indicate an increased level of arousal in depression. 

#### 3.2.2. Hyperactivity of HPA Axis

Hyperactivity of the HPA axis in MDD has been broadly reported [22,23,90]. The hypothalamus secretes corticotrophin-releasing factor (CRF) and vasopressin, which in turn activate the pituitary to release adrenocorticotropin hormone (ACTH). This finally stimulates the release of cortisol from the adrenal cortex [23]. Typically, hyperactivity of the HPA axis is always indirectly detected in humans by measuring hormone levels such as cortisol and ACTH. Many studies have reported elevated cortisol levels in the plasma [91,92,93,94,95,96] and urine [97,98] of depression patients. For instance, the mean plasma cortisol level of depressed patients before treatment was observed to be elevated by 10 μg (per 100 mL) above normal levels [91]. An analysis of plasma cortisol every 20 min over a 24-h period revealed higher cortisol levels and more frequent secretory episodes in depressed patients compared to normal individuals [92]. The mean urinary free cortisol level in depressed patients was significantly elevated (90.1 μg), compared to the normal level (48 μg) [97]. It has also been observed that cortisol levels can return to their normal levels following treatments like electroconvulsive therapy or antidepressive drugs (imipramine) [91,98]. Notably, cortisol levels in MDD patients, both during depressed and recovery states, are higher than in healthy individuals [99]. A meta-analysis further supports these findings, indicating a robust elevated ACTH level in depression [22]. The increase in the ACTH level in MDD after administration of CRF is slower than in healthy individuals [99,100]. Additionally, research indicates that the volume of the adrenal gland increases during depressive episodes in MDD and returns to normal size during remission [101].

#### 3.2.3. Hyperactivity of Noradrenergic System (LC)

In MDD, hyperactivity of the central noradrenergic system, particularly the LC, is evident. Specifically, there is a notably higher level of norepinephrine (NE) and its metabolites. NE levels in the cerebrospinal fluid of MDD patients during 30-h recordings are consistently higher compared to healthy individuals [102]. Additionally, the appearance of NE in both extravascular and vascular compartments is elevated in MDD compared to healthy individuals [103]. Furthermore, the concentration of noradrenaline metabolites is also higher in the saliva of MDD compared to healthy individuals [104]. 

Hyperactivity of the LC–NE system in MDD is also reflected in the abnormal activities of neurotransmitters, including glutamate and tyrosine hydroxylase. It has been reported that the LC primarily receives excitatory input from glutamate and increased glutamatergic activity in MDD. Specifically, increased gene expression of glutamate receptors and a deficiency in astrocyte glutamate transporter gene expression have been observed in MDD post-mortem studies [105,106]. Furthermore, levels of tyrosine hydroxylase, which reflect the neuronal activity of the LC, have been found to be elevated in the LC of post-mortem MDD brains [107].

Moreover, the effectiveness of some antidepressants is associated with down-regulation of the LC’s activity. For instance, the selective NE reuptake inhibitor (reboxetine) has been shown to reduce the firing activity of NE neurons in the LC of rats [41,108]. This effect is also observed following a series of electroconvulsive shocks [109]. These findings suggest a close link between the pathogenesis of depression and the hyperactivity of the central noradrenergic system, potentially contributing to the hyperarousal pattern in depression.

#### 3.2.4. Hyperstable Arousal Regulation as Indexed by EEG Vigilance

Electrophysiological evidence, characterized by EEG vigilance markers such as alpha, theta, and delta activity, demonstrates a typical hyperstable and higher arousal state in MDD patients than healthy individuals [110]. During resting states without external interruptions, most healthy individuals exhibit a progressive decline to a lower vigilance stage, suggesting a decrease in arousal levels [3]. In contrast, MDD patients often exhibit a hyperstable pattern of arousal regulation [3,4,5]. 

For instance, in a 15-minute EEG recording with closed eyes, unmedicated MDD patients spend more time in the highest EEG vigilance stages and exhibit a delayed decline to lower EEG vigilance stages compared to healthy individuals [4]. Subsequent studies have consistently confirmed the hyperstable arousal regulation pattern in depressed patients, especially in MDD [6,110,111,112,113,114,115]. Notably, even during two-minute EEG recordings, this pattern is observed [6]. It has also been found that more sleep disturbances before the day of recording correlate with a higher score of arousal stability in depression patients, but not in healthy individuals [114]. Similarly, symptom severity measured using the Beck Depression Inventory (BDI) was found to correspond with higher arousal levels and slower declines in arousal as indexed by EEG vigilance [113]. And patients with bipolar disorder in a depressive episode also exhibit higher mean vigilance levels as measured by EEG vigilance [116].

Moreover, studies have revealed that the arousal regulation pattern was related with responses to antidepressant treatments [111,115]. Specifically, during 15-minute EEG recordings, individuals who responded to antidepressants (like escitalopram or/and mirtazapine) exhibited a reduction in time spent in high vigilance states and an increase in time spent in low vigilance states two weeks after beginning treatment, compared to those who did not respond [111]. Remitters demonstrate a stronger tendency to decline to lower arousal levels compared to non-remitters [115]. 

In summary, evidence for abnormal hyperarousal in depressed patients is supported by a range of behavioral and physiological findings. These include sleep disturbances, increased heart rate, enlarged pupil diameters, heightened skin conductance, and hyperactivity in the subcortical areas, particularly in the hypothalamus and LC (Figure 2). Recent EEG studies suggest a hyperstable arousal regulation in depressed individuals that is linked to the severity of depressive symptoms. 

Despite the comprehensive results these studies provide, several limitations must be acknowledged. Firstly, although the studies suggest correlations between arousal regulation patterns and antidepressant treatments, the causality cannot be firmly established due to their observational nature. Additionally, the distinction between neurobiological underpinnings of hyperarousal in MDD and in other mood disorders, such as the depression phase in bipolar disorder and post-traumatic stress disorder [76,116,117,118,119], require further investigation. More importantly, previous studies did not thoroughly explore the association between subcortical–cortical connections, particularly thalamocortical interactions, and abnormal arousal in depression. The neural mechanism underlying various depressive symptoms—phenomena that involve multiple cognitive processes and cortical activities [120]—as responses to abnormal hyperarousal in depression remains poorly understood.

## 4. Thalamocortical Circuits Possibly Account for Abnormal Arousal in Depression

Thalamocortical circuits have been implicated in the regulation of physiological arousal [121,122]. The central thalamus is specialized to maintain the thalamocortical and cortico-cortical connections, prompting brief shifts in arousal, and injury to the central thalamus induces impairment of arousal regulation [123]. The firing of centromedial thalamus neurons has been proposed to implement dual control over sleep–wake states by modulating brain-wide cortical activities [124]. During non-rapid eye movement (NREM) sleep, the thalamus has also been found to modulate the slow waves that predominate in the neocortex [125]. Recent studies in macaques have shown that deep brain stimulation of the central or central lateral thalamus can facilitate their interactions with the brain-wide cortex, thereby restoring arousal from an anesthetized state [59,73]. Additionally, human research using fast fMRI has revealed a temporal sequence of activity across thalamic nuclei and the brain-wide cortex during the transition in arousal, suggesting thalamocortical dynamics that support arousal [126]. These studies provide empirical evidence for the critical role of thalamocortical circuits in the arousal system. 

In fact, the thalamus and its cortical connections exhibit functional aberrance in patients with depression. Previous research has identified a positive correlation between increased thalamic metabolism and the severity of depressive symptoms [127,128], while a decrease in thalamic metabolism has been observed during remission in depressed patients [129]. Furthermore, several studies have revealed altered thalamic connectivities in depression, including increased thalamic connectivities with the default-mode network [130,131], somatosensory cortex [132,133], temporal cortex [133], insula [134], and dorsolateral prefrontal cortex [135], as well as decreased thalamic connectivities with the anterior cingulate [130,131]. Importantly, thalamocortical interactions appear to play a critical role in the pathology of depression. A recent review demonstrated converging evidence of enhanced effective connections from the thalamus to various cortical regions, and reduced effective connections from other regions to the thalamus, suggesting that the thalamus is the key casual hub region for MDD [136]. Moreover, a study utilizing machine learning and advanced deep learning methods to distinguish MDD patients and healthy individuals in large resting-state fMRI datasets, identified thalamocortical hyperconnectivity as a specialized and critical neurophysiological signature in MDD [137]. Given the importance of thalamocortical circuits in arousal regulation, as previously discussed, it is proposed that hyperconnectivity of this circuit could be the subcortical–cortical neural mechanism underlying abnormal arousal regulation in depressed patients.

Except for the thalamus, several subcortical regions are related to arousal, such as the brainstem, basal forebrain, and hypothalamus, but few of them have been found to show altered connectivity in the cortex of depressed patients. Based on various evidence regarding aberrant thalamocortical connections in depression, it can be inferred that hyperconnectivity of thalamocortical circuits may play a significant role in hyperarousal and depressive symptoms (Figure 3).

## 5. Future Directions

Currently, converging findings have revealed hyperarousal in depression, as indicated by sleep difficulties [8,77,78], increase in HR [79,80,81,82,83,84,85,86,87], decrease in HRV [88,89], pupil diameters and skin conductance [87], and hyperactivity of the HPA axis [22,23,90,91,92,93,94,95,96,97,98,99,100,101] and locus coeruleus [102,103,104,105,106,107] in depressed individuals. Given the intimate involvement of thalamocortical circuits in physiological arousal, it is proposed that thalamocortical circuits could account for abnormal arousal regulation in depression. However, existing research has not yet elucidated how these abnormalities in thalamocortical circuits lead to hyperarousal regulation in depressed individuals. To further elucidate this relationship, advanced techniques such as simultaneous fMRI–EEG or fMRI–pupillometry could be further utilized to investigate the dynamics of thalamocortical interactions and their impact on arousal regulation in depression.

The thalamus possesses a complex neuroanatomy, comprising a variety of nuclei and including both excitatory and inhibitory neurons. Its connections to other brain areas are diverse, contributing to a wide range of cognitive and behavioral functions [138]. For instance, based on histological criteria, the thalamus is divided into distinctive nuclei, such as centromedial, ventral posterolateral, pulvinar, etc. [139]. Moreover, according to the patterns of afferent connection, excitatory thalamic nuclei could be divided into first-order nuclei (e.g., lateral geniculate nucleus), which receive driver input from subcortical regions, and higher-order nuclei (e.g., mediodorsal and pulvinar nuclei), which receive modulatory inputs from the cortex [138,140]. This classification, however, varies depending on the criteria used [139], suggesting the complex anatomy of thalamus and its cortical connections. A recent study found that, compared to normal controls, first-episode, drug-naïve MDD patients exhibit increased gray matter volume in specific thalamic nuclei, but not in the whole thalamus, suggesting heterogeneous alterations across thalamic nuclei [141]. Therefore, it is crucial to further investigate the interactions between various thalamic nuclei and the cortex in depressed individuals [142], in order to determine which of these interactions is related to abnormal arousal in depression.

Using rTMS that targets the dorsolateral prefrontal cortex (DLPFC) is a common non-pharmacological clinical treatment for MDD. However, recent studies indicate its limited effectiveness, with less than half of treatment-resistant MDD patients responding to DLPFC-rTMS [143]. Additionally, rTMS at other alternative targets near the DLPFC, such as the dorsomedial prefrontal cortex, did not shown improvements in depressive symptoms for treatment-refractory depression [144]. This underscores the urgent need to identify alternative rTMS targets, particularly for those with treatment-resistant depression. Intriguingly, rTMS at the left motor cortex, which has been proposed to be intimately associated with arousal systems [72], has recently been found to have comparable efficacy to DLPFC stimulation in MDD patients who show psychomotor retardation [145]. Given this context, we propose that the cortical regions connected to the thalamus are related to hyperarousal regulation in depression, and may be potential targets for rTMS in treating MDD patients.

Moreover, music intervention is increasingly used in depressed symptoms alleviation. Previous study has shown the impact that music has on stress levels in healthy individuals [146]. A recent meta-analysis investigating the effects of music interventions on depression confirmed their efficacy, both in terms of music medicine or music therapy [147]. Future research should explore whether the mechanisms underlying the effectiveness of music intervention are linked to arousal modulation, and can thereby contribute to the alleviation of depressive symptoms. This exploration could guide the development of more effective, non-invasive treatments.

In addition to non-invasive approaches, pharmacotherapy is another effective means of depression treatment. Recent studies have revealed ketamine’s effectiveness as an antidepressant in patients with treatment-resistant depression [148,149,150,151]. It should be noticed that, ketamine, commonly used as an anesthetic, is known to reduce arousal levels [152,153,154,155,156]. However, it remains unclear whether its antidepressant effects are linked to a reduction in thalamocortical connectivity in MDD. Future research is needed to determine whether ketamine alters thalamocortical interactions, and how such changes may contribute to its antidepressant effects. Should a link between thalamocortical connectivity and antidepressant effects be established, it could shed light on the development of targeted drug treatments in the future by focusing on their impact on thalamocortical connectivity.

Recent findings suggest that sex differences [157] should be considered in the neural mechanisms underlying the drug treatment. Specifically, hyperarousal in MDD exhibits sex differences [157], with women often experiencing more hyperarousal symptoms compared to men, a condition linked to excessive secretion of CRF [158,159]. Females show increased sensitivity to CRF in the LC due to sex differences in the CRF1 receptor, leading to increased cyclic adenosine monophosphate–protein kinase A (cyclic AMP–PKA) pathway signaling [160,161,162]. This increased response may contribute to increased hyperarousal symptoms in females [163,164,165]. In contrast, the CRF1 receptor in males preferentially binds to β-arrestin, leading to different signaling pathways and potentially mitigating hyperarousal in the context of CRF hypersecretion [160,166,167,168]. These findings indicate that sex-specific responses to stress and CRF regulation could promote the development of targeted drug treatments for MDD that take into account these sex differences.

## 6. Conclusions

In conclusion, converging evidence indicates a hyperarousal pattern in depressed patients, characterized by sleep disturbance, increased arousal-related biological markers, hyperactivity in the HPA axis and LC, and hyperstable EEG vigilance. Notably, there have been limited studies investigating the contribution of thalamocortical circuits to abnormal arousal in depression. This knowledge gap hinders our understanding of the neurobiological underpinnings of how abnormal arousal contributes to various depressive symptoms, which are phenomena involving complex cognitive functions and multiple cortical regions. By examining the critical role of thalamocortical connections in arousal regulation and MDD, we propose that hyperconnectivity of thalamocortical circuits could account for both the hyperarousal pattern and the related social difficulties in depressed patients. Future investigations should adopt advanced techniques, such as simultaneous fMRI–EEG or fMRI–pupillometry, to elucidate the relationship between thalamocortical interactions and hyperarousal in depression, specifically identifying which nuclei and their cortical interactions contribute to abnormal arousal. This could provide potential targets for rTMS treatment. As for other non-invasive treatments, music intervention shows promise; however, its mechanisms and the potential links with arousal need further investigation. In pharmacotherapy, the development of targeted drugs in treatment-resistant depression could consider modulating thalamocortical connectivity. Additionally, observed sex differences in the neurobiological underpinnings of hyperarousal in MDD should be considered in future hyperarousal and treatment research.

## Figures and Tables

**Figure 1 brainsci-14-00050-f001:**
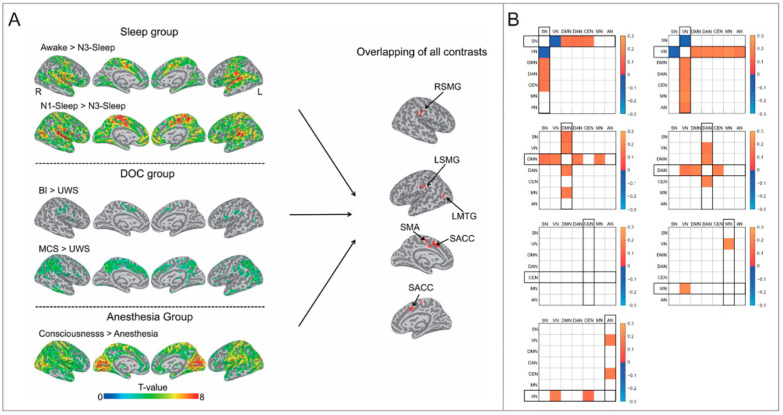
Evidence of altered cortical connectivity in different arousal levels. (**A**) The brain regions with decreased degree centrality during unconsciousness. Adapted from [69]. SMA = supplementary motor area; LSMG = left supramarginal gyrus; RSMG = right supramarginal gyrus; LMTG = left middle temporal gyrus; SACC = supragenual anterior cingulate cortex. (**B**) The FC differential patterns (EC–EO) between networks. Adapted from [73]. FC = functional connectivity; EC = eye closed; EO = eye open; SN = salience network; VN = visual network; DMN = default mode network; DAN = dorsal attention network; CEN = central executive network; MN = motor network; AN = auditory network.

**Figure 2 brainsci-14-00050-f002:**
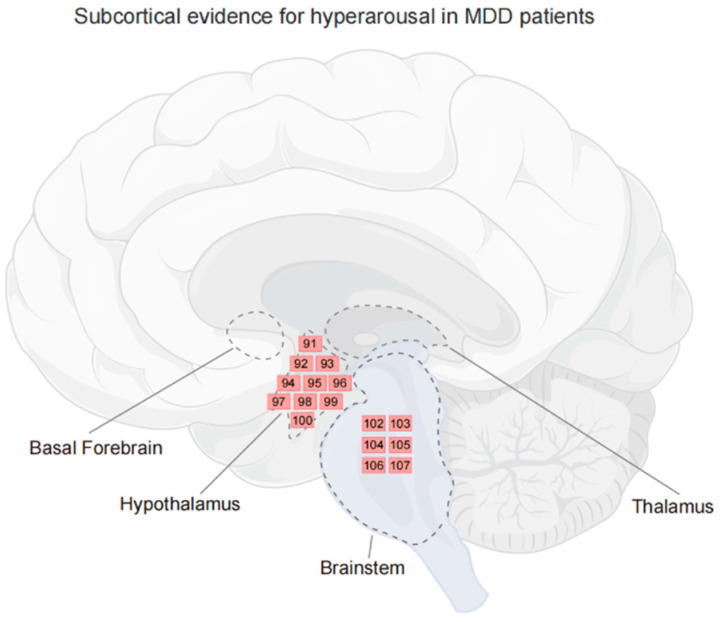
Subcortical evidence for hyperarousal in MDD patients. Regions where activity is related to the hyperarousal in MDD. Each number corresponds to a reference, and indicates the contribution of the region, not to an exact location. The subcortical areas related to arousal are delineated by dashed lines. Figure created using FigDraw.

**Figure 3 brainsci-14-00050-f003:**
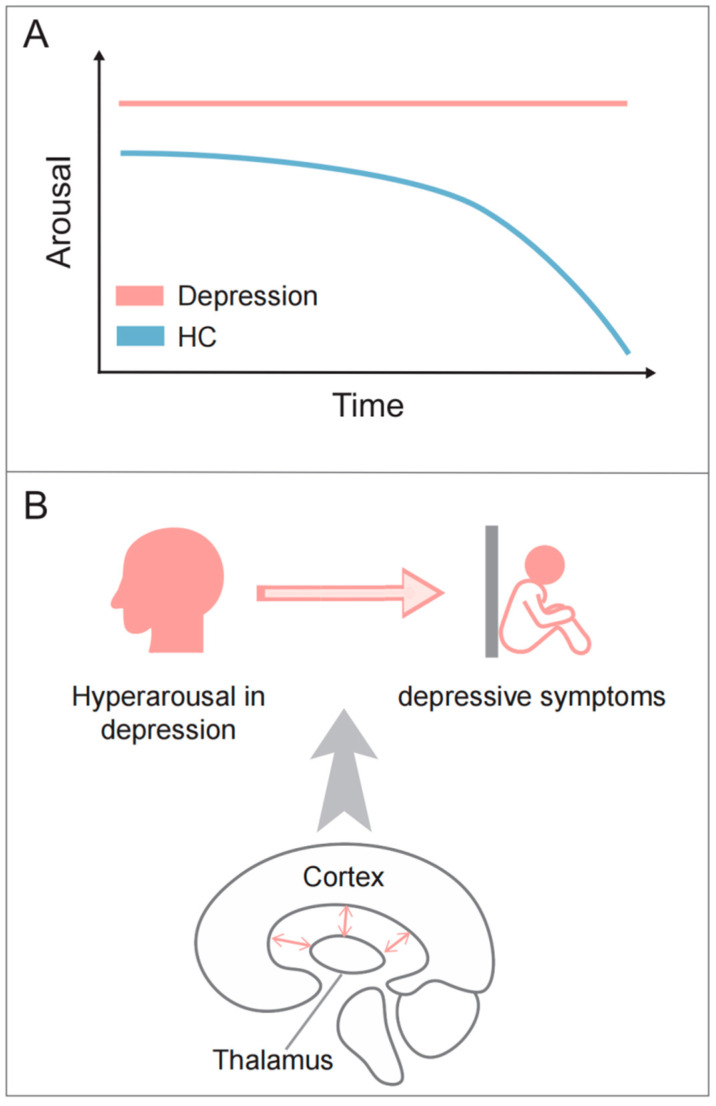
The abnormal arousal pattern in depression. (**A**) During resting state, healthy individuals exhibit a progressive decline to lower arousal, while MDD patients often exhibit a hyperstable pattern of arousal regulation; (**B**) thalamocortical circuits may contribute to both the hyperarousal pattern and depressive symptoms.

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
