# Peer review of "Neurobiological Underpinnings of Hyperarousal in Depression: A Comprehensive Review"

_brainsci, 2024, doi:10.3390/brainsci14010050_

Round 1

Reviewer 1 Report

Comments and Suggestions for Authors

This paper holds relevance in the context of understanding major depressive disorder (MDD) by delving into the lesser-explored aspect of abnormal arousal. It provides a comprehensive overview of the physiological, neural, and behavioral markers associated with arousal dysregulation in MDD, highlighting its potential as a critical factor in depressive symptoms.

The paper's primary objective is to elucidate the neurobiological underpinnings linking abnormal arousal to depressive symptoms. It systematically presents evidence of hyperarousal in depression and suggests the thalamocortical circuit as a key neural mechanism influencing both arousal patterns and depressive manifestations. The proposed focus on this circuit's role in modulating depressive symptoms is a notable strength, indicating the paper's potential contribution to understanding MDD.

Through an extensive literature review, the paper synthesizes existing findings effectively. It convincingly outlines the physiological and neural markers of hyperarousal in depression, linking them to various behavioral manifestations observed in depressed individuals. The proposal of thalamocortical circuit involvement in both hyperarousal and depressive symptoms is thought-provoking, opening avenues for further investigation.

While comprehensive, methodological transparency needs to be be improved by explicitly detailing the criteria for literature selection and synthesis methodologies. And this is my only major comment for article revision.

Reviewer 2 Report

Comments and Suggestions for Authors

Undoubtedly, patients with major depressive disorder exhibit abnormal physiological arousal pattern, known as hyperarousal, which may contribute to their depressive symptoms. 

Authors in this paper overview the physiological and neural features of arousal, and reviews the literature indicating abnormal arousal in depressed patients. 

My comments on the article are as follows:

- I propose to expand the introduction to the article by referring to research on the impact of music on humans, based on EEG. For example, you may refer to: The Impact of Different Sounds on Stress Level in the Context of the EEG, Cardiac Measures and Subjective Stress Level: A Pilot Study, Brain Sciences 10(10), 728, 10.3390/brainsci10100728.

- I also suggest expanding the range of keywords, because only three of them are presented, and there are definitely more such words in the article.

- Subchapters in section: 3. Abnormal arousal in Depression should discuss the issues in more detail. The current description is sparse.

- The Summary should discuss possible future plans in more detail. Currently, this is only partially compensated by the chapter: 5. Future direction

Comments on the Quality of English Language

Minor editing of English language required. 

Reviewer 3 Report

Comments and Suggestions for Authors

The manuscript reports an interesting review of the existing literature about neurobiological features of depression. The paper is clear, well-written, and reports the relevant evidence of the existing literature. I have only a few comments for the authors, to improve their manuscript:

- have you evaluated a possible paragraph about the presence of specific limits in the current literature that might have reduced the current knowledge?

- have you evaluated the possible role of anterior cingulate cortex, fusiform gyrus and insula as filter of arousal that could compromise the response to stimuli (see 10.1186/s40359-014-0052-1)?

- I think at this time gender differences have to be considered. Are there any elements that you could include in your paper about this aspect? Clinical data have reported the presence of different symptomatology mediated by gender and this might be included - as well as different responses to drugs.

- future directions paragraph is focused on biological treatments. Is there any space for new drug targets?

Round 2

Reviewer 1 Report

Comments and Suggestions for Authors

Thank you very much for considering my comments.

Reviewer 2 Report

Comments and Suggestions for Authors

The changes have been implemented.